# Optical Sensing of Toxic Cyanide Anions Using Noble Metal Nanomaterials

**DOI:** 10.3390/nano13020290

**Published:** 2023-01-10

**Authors:** Ramar Rajamanikandan, Kandasamy Sasikumar, Saikiran Kosame, Heongkyu Ju

**Affiliations:** Department of Physics, Gachon University, Seongnam-si 13120, Republic of Korea

**Keywords:** water toxicity, cyanide sensing, noble metal nanomaterials, surface plasmon resonance, fluorescence, surface-enhanced Raman scattering, Rayleigh scattering

## Abstract

Water toxicity, one of the major concerns for ecosystems and the health of humanity, is usually attributed to inorganic anions-induced contamination. Particularly, cyanide ions are considered one of the most harmful elements required to be monitored in water. The need for cyanide sensing and monitoring has tempted the development of sensing technologies without highly sophisticated instruments or highly skilled operations for the objective of in-situ monitoring. Recent decades have witnessed the growth of noble metal nanomaterials-based sensors for detecting cyanide ions quantitatively as nanoscience and nanotechnologies advance to allow nanoscale-inherent physicochemical properties to be exploited for sensing performance. Particularly, noble metal nanostructure e-based optical sensors have permitted cyanide ions of nanomolar levels, or even lower, to be detectable. This capability lends itself to analytical application in the quantitative detection of harmful elements in environmental water samples. This review covers the noble metal nanomaterials-based sensors for cyanide ions detection developed in a variety of approaches, such as those based on colorimetry, fluorescence, Rayleigh scattering (RS), and surface-enhanced Raman scattering (SERS). Additionally, major challenges associated with these nano-platforms are also addressed, while future perspectives are given with directions towards resolving these issues.

## 1. Introduction

A cyanide ion has the acutest toxicity among inorganic anions and needs to be monitored due to its fatally harmful effects on ecosystems involving humans. This chemical is deadly poisonous with particular regard to the respiratory, cardiovascular, and nervous systems of mammals, including humans, leading to the death of living organs after exposure at a low dose for only a few minutes [1,2,3]. It comprises a carbon atom (C) triple-bonded with a nitrogen atom (N) through a negative charge (C≡N^−^). Cyanides that are naturally found both in the biological and geological world are produced by many plants as a spontaneous shield against pests [4,5,6,7,8]. However, industrial activities discharge exceedingly vast amounts of cyanides, such as in metallurgy, electroplating, manufacturing of plastics/organic reagents, mining, fumigation, and photographic development [9].

The default guideline values to protect drinking water have been set to (76 nM) 2 μg L^−1^ and (0.15 μM) 4 μg L^−1^, in the New Zealand and Australian Environmental and Conservation Councils, respectively [10,11]. The United State Environmental Protection Agency (USEPA) has set the highest allowable amounts of cyanides in drinking water at (7.6 μM) 200 μg L^−1^ [12]. The European Union has set a lower level of (1.9 μM) 50 μg L^−1^ of cyanides in environmental water [13]. Furthermore, the Drinking Water Management Act in The Republic of Korea has set the level for cyanides at less than (0.39 μM) 10 μg L^−1^ as the drinking water standard [14]. With these guidelines concerning cyanides, ecosystem protection urgency has led to the development of sensor devices capable of quantitatively detecting it, with a detection limit of sub-μg L^−1^ in water. Several analytical sensing technologies have been developed for the detection of cyanide anions, such as those based on electrochemistry, chemiluminescence, titration, potentiometry, voltammetry, fluorescence/colorimetry with organic molecules and metal complexes, gel-based visual detection, etc. [1,2,9,15,16,17,18,19,20]. Several review articles have been published for the wide range of methodologies developed for cyanides detection [4], such as for the fluorometer- and colorimeter-based detection with organic molecules and metal complexes [5] and the gel-based visual detection of cyanides [3]. 

Among the methodologies reported, the optical chemosensors for cyanides detection, such as those based on colorimetry, fluorescence, Rayleigh scattering (RS), and surface-enhanced Raman scattering (SERS), have gathered continuous attention due to their advantages, including convenient and/or simple operation, miniaturized device fabrication, inexpensive sensing operation, high sensitivity with adequate specificity and real-time detection capability [21,22,23,24,25]. 

Advances in nanotechnologies and nanoscience have allowed the fabrication of various nanostructured materials with intrinsic morphologies, such as the nanoparticles (NPs) of copper/silver/gold (CuNPs/AgNPs/AuNPs) [26,27,28,29,30,31,32,33,34,35], and the nanoclusters (NCs) of the metals (CuNCs/AgNCs/AuNCs) [30,31]. For sensing strategies, analyte-induced changes of the physicochemical properties near those nanoparticles are designed to alter their opto-chemical response. Such local changes include the occurrence of conglomeration/anti-conglomeration [27,28], morphology modification [17,19], optical refractive index change [36,37,38,39], and the introduction of external dipole coupling [39,40]. 

Metal nanoparticles have been involved in optical sensing platforms, including those based on colorimetry [27,28], surface plasmon resonance [41,42,43,44], fluorescence [45,46,47], RS [48], and SERS tactics [49,50]. These sensing devices have produced reasonably good sensitivity in detecting analytes with adequate specificity, whereas the high reproducibility of the sensor signals remains a challenge due to limited control over the uniformity of nanoparticle size to date. The fact that the metals of the nanomaterials chemically react with cyanides through the so-called Elsner reactions can be exploited to design optical chemosensors that can detect cyanides quantitatively [16,17]. This chemical reaction, that produces metal-cyano complexes in an oxygenated environment, can be combined with the nanoparticle properties that govern optical properties, such as resonance spectroscopy, fluorescence, and elastic/inelastic light-scattering spectroscopy. 

This paper focuses on the review of the metal nanostructure-based optical chemosensors that have been developed for the analytical detection of cyanides, such as those utilizing the plasmonic MNPs-based colorimetry, plasmonic MNPs-enhanced elastic/inelastic scattering (RS and SERS), and metal nanoclusters (NCs)-based fluorescence sensors. Plasmonic colorimetry uses the morphological changes of MNPs in the particle accretion that follows the cyanide-metal interaction. The fluorescence-based nanosensors for detecting cyanide ions are reviewed, with approaches that employ cyanide-induced effects such as fluorescent probe etching, metal-cyano complex formation, inner-filter effect (IFE), and Förster resonance energy transfer (FRET). RS and SERS approaches with plasmonic MNPs are reviewed, with a focus on the quantitative detection of cyanide through NPs conglomeration. Lastly, we offer the conclusions of the review with an outlook for noble MNPs-based optical chemosensors for cyanide detection. Figure 1.

## 2. Plasmonic Nanomaterials-Based Colorimetric Sensors for Cyanide Ions

Colorimetry-based sensors make use of color changes caused by the development of physicochemical assemblies, morphological changes, and chemical reactions [49]. Generally, the quantification of various potential analytes was successfully accomplished using a variety of organic and biological colorimetric probes. However, the use of organic and biological colorimetric probes was limited by frequently challenging synthesis procedures, difficult fabrication processes, expensive/highly technical determination techniques, and the requirement for extremely expensive chemical reagents [3,5,51,52]. Therefore, easy, realistic, selective, and sensitive colorimetric probes for the quantization of toxic/biological analytes from problematic samples are required. To overcome this issue, plasmonic nanomaterials-based colorimetric sensing analysis for potentially important toxic and biological analytes is very interesting, due to the simple preparation and easy operation [51,52,53,54,55]. 

### 2.1. AgNPs-Based Colorimetric Sensors for Cyanide Ions

With surface plasmon resonance, AgNPs exhibit optical absorbance coefficients larger than other MNPs of a similar size and shape due to the smaller magnitude of the imaginary component of the permittivity of silver, combined with a sufficiently large magnitude of its real component (as a negative value). This feature leads to a narrower peak in the plasmon-induced optical absorbance spectrum with a higher quality (Q) factor, favoring higher sensitivity for visual color changes [56,57,58,59]. 

S. Hajizadeh et al. [60] reported the high sensitivity and specificity for sensing cyanide ions in water using sodium dodecyl sulfate (SDS) -functionalized AgNPs with naked-eye visual observation. The presence of the cyanide ions caused the plasmonic optical response of SDS-AgNPs to be reduced, in turn decreasing the color contrast, resulting in a colorless state. This colorimetric sensor showed the linear relationship between plasmonic absorbance and the cyanide ion concentration in the range of 16.7–133.3 μM at the wavelength 394 nm. The SDS-AgNP-based approach obtained the cyanide detection limit of 1.8 μM and was exploited for its application in cyanide detection in dam water samples. 

N. Pourreza and co-workers [61] presented an approach for eco-friendly, in-situ synthesis of AgNPs embedded in flexible and transparent bacterial cellulose nanopapers for cyanide ion detection. In this technique, adsorbed Ag^+^ ions on bacterial nanofibers were reduced by the hydroxyl radical of cellulose nanopapers; the reducing agent generated the bio-nanocomposite, i.e., the embedded AgNPs in transparent nanopapers (ESNPs). ESNPs allowed incremental amounts of cyanide ions to cause a strong hypsochromic shift in the absorbance peak. This appeared to be due to cyanide ion-induced etching of the AgNP surface; the subsequent alteration of the plasmon resonance resulted in the shift. This sensor was further applied to real water samples, such as those from industry and the environment. 

H. Hatam et al. [62] reported the super specific and highly sensitive colorimetric method for quantifying cyanide ions in environmental water samples using plasmonic AgNPs embedded in a transparent agarose matrix. This sensor displayed a wide linear range of the signal versus cyanide ion concentrations from 1.5 to 120 μM, with a detection limit of 0.69 μM. This sensor also presented remarkable selectivity for detecting cyanide ions against various other inorganic anions of concentrations exceeding 50 times that of cyanide ions. 

K.F. Princy et al. [63] studied the colorimetric and fluorescent turn-on sensing of cyanide ions in a water medium using seaweed (marine macroalgae) facilitated biosynthesized AgNPs. The bioactive substances in the seaweed extract acted as reducing agents as well as stabilizing agents for AgNPs. Insertion of cyanide ions caused the color of the seaweed AgNPs colloid to change from brown to colorless due to a decrease of plasmonic absorbance at the wavelength of 414 nm with concomitant fluorescence. This fluorescence turned on as the size of AgNPs became small enough to reach the Fermi wavelength of an electron (0.5 nm), thus being referred to as AgNCs that acted as fluorescent probes. This technique was used to detect cyanide ions in environmental water samples with its device integrated with a test strip for real-time sensing of cyanide ions. 

P. Paul and co-workers [64] synthesized the rosmarinic acid AgNPs for the colorimetric sensing of cyanide ions in environmental water samples. The redox reaction that took place on the surface of the nanoparticles, which was mediated by cyanide ions and reduced dissolved oxygen into ROS (O_2_ to O_2_^−^, O_2_^2−^ etc.), and oxidized silver (Ag^0^) into Ag^+^, was the subject of the sensing mechanistic pathway that was investigated by this nanosensor. Thus, Ag^+^ reacted with cyanide to form a water-soluble AgCN complex and with O_2_^−^ establishing Ag_2_O. This annihilation of AgNPs triggered a color alteration from yellow to colorless, with fading of the plasmonic absorbance peak (band). For portable usage, agarose-based test strips were merged by arresting the rosmarinic acid-AgNPs onto the agarose test strip and was effectively utilized for the determination of cyanide ions in a water medium. The majority of the AgNP-based cyanide ion chemosensors may have been altered by the chemical interaction between the cyanide ions and AgNPs in an oxygenated environment (Figure 1) [65,66]. In addition, AgNP-based colorimetric detection of cyanide anions is summarized in Table 1, showing that most of the methodologies obtained an LOD of less than or almost equal to that permitted by WHO regarding cyanide levels in drinking water. 

### 2.2. AuNP-Based Colorimetric Sensors for Cyanide Ions

AuNPs have been widely used in several scientific fields in recent decades, especially analytical chemistry, making them one of the most widely accepted nanostructured materials [67,68]. Due to their high extinction coefficient and interparticle interaction based SPR band, AuNPs are the appropriate material for colorimetric nanoprobes for the optical detection of cyanide anions in entirely water-based settings. According to A. Pal et al. [69], pink-colored stable AuNPs were produced photochemically in an aqueous Triton X-100 (TX-100) solution and used for spectrophotometric cyanide anion quantification. TX-100-AuNPs’ colorimetric properties in combination with their SPR band cause cyanide addition to be reduced; at the endpoint of the experiment, it becomes colorless. This sensor was used to find cyanide anions in samples of industrial effluent. 

Adenosine triphosphate (ATP)-functionalized AuNPs with copper-phenanthroline ensemble-based naked-eye visual sensing of cyanide anion was shown by M.H. Kim and colleagues [70]. With the ligand-exchange method used by this sensor, a chemical change results from the replacement of one ligand with another. As a result, cyanide interacts with AuNPs in the presence of copper-phenanthroline, which may disrupt AuNPs dispersion. Due to the cyanide anion’s chemical reaction with this nanoprobe, the SPR band and colorimetric response were observed. The W.-L. Tseng research team [71] described using polysorbate-40 (PS-40) coated AuNPs to colorimetrically detect cyanide, endogenous biological cyanide, and hydrogen cyanide in plants harboring cyanogenic glycosides. The cyanide anion was successfully detected in loquat kernels, cassava roots, and peach kernels using this nanosensor. Chitosan-AuNP film was used by C. Radhakumary’s research group to assess cyanide ions in biological (blood) and water samples [72]. Similar to the colloidal chitosan-AuNP solution, this chitosan-AuNPs film had an SPR band at 534 nm. Surprisingly, this nanosensor provides for quick, naked-eye detection of cyanide ions at levels equivalent to or higher than 2 μg L^−1^. In addition, the researchers claim that their nanosensor was a low-cost, portable, and simple-to-use method for the on-site monitoring of actual water and biological samples, commonly used in underdeveloped nations where access to sophisticated apparatus is constrained. 

In order to produce a quick and simple colorimetric detection of cyanide anions in real water samples, Ilanchelian, et al. [73] employed AuNPs functionalized with β-cyclodextrin (β-CD). This optical chemosensor has a detection limit of roughly 93 nM, and a broad linear range between 4.5 and 99 μM. This approach also included a highly stable cotton swab for user convenience. β-CD Cotton swabs coated with AuNPs display good sensitivity to the cyanide anion. When a cotton swab covered with β-CD-AuNPs was examined under a scanning electron microscope and then immersed in a 99 μM solution of cyanide, its color rapidly changed from wine red to colorless (Figure 2). Using stimuli-responsive poly (N,N-dimethylamino ethyl methacrylate (PDA) modified AuNPs, Z. Alinejad, et al. [74] demonstrated a simple naked-eye visual method for the detection of cyanide anion in a wide linear range. However, this nanosensor worked best at pH = 9. PDA-AuNPs easily interact with cyanide anions through chemical interaction, being decomposed into smaller particles. A drop in the SPR peak at 530 nm and a shift in the colloidal dispersion’s color from wine-red to colorless were reliable indicators that the chemical reaction was being followed effectively using colorimetric reflection.

M. Budlayan and colleagues [75] discovered the potential colorimetric detection of cyanide anion in natural water samples. Here, PVA-chitosan-coated AuNPs were used to produce a thin film. When the cyanide ions reacted with the AuNP coating, the thin film’s color changed from red to colorless. The naked eye could see it well. This nanosensor provides a strong substitute for a quick, label-free, and generous approach to detecting cyanide for environmental on-site monitoring and water pollution mitigation. The same Elsner reaction is followed by all AuNP-based sensing techniques (Figure 3e). Based on this Elsner reaction, numerous research teams created compounds that functionalized AuNPs and used them to quantify the amount of cyanide in ecological water samples [76,77,78]. Typically, UV-vis absorbance, HR-TEM, colorimetric images, and mass spectral data have been employed in cyanide detection procedures (Figure 3).

### 2.3. CuNP-Based Colorimetric Sensors for Cyanide Ions

In comparison with other precious metals like Ag and AuNPs, CuNPs have significant electrical, optical, and conductivity properties. The main problem with CuNPs is that they are quickly oxidized, resulting in copper oxide nanoparticles. Similar to Ag and AuNPs, CuNPs equally display the SPR absorption peak in the visible area [29,80]. CuNPs are produced cost-effectively and are widely utilized in the SPR-linked colorimetric detection of cyanide anions. In situ production of an inventive bio-nano composite with “embedded CuNPs in nanocellulose film (ECNPs-NC)” was described by Pouzesh, et al. [81] and used in the detection of cyanide. The synthetic bio-nano composite (ECNPs-NC) film offers greater promise as a practical and affordable optical chemosensor for measuring the level of very deadly cyanide anions in water samples. The SPR band intensity varied linearly with changes in cyanide anion concentration, with an LOD of 0.015 μg mL^−1^ and a range of 0.25 to 0.40 μg mL^−1^. This method is efficient, but it has several drawbacks, including the lengthy process that is required due to multiple preparatory steps, challenging processes, and the necessity for issues with monitoring the uniform size distribution of the CuNPs in the NC film. During surface functionalization with various molecules, AuNPs exhibit the highest stability amid other nanoparticles such as AgNPs and CuNPs [49,67,68]. Moreover, such stability of AgNPs appears to be higher than that of CuNPs.

### 2.4. Core–Shell Nanoparticles-Based Colorimetric Sensors for Cyanide Ions

To date, using bimetallic plasmonic core–shell nanostructured materials, cyanide can be detected with high sensitivity in a simple sensing format [82]. The SPR absorbance peak of plasmonic core–shell NPs will alter due to changes in the shell–to–core width relation caused by cyanide anion. According to Zeng, et al. [83], cyanide anion detection utilizing Au@Ag core–shell NPs provides a sensitive colorimetric platform. The ratio of the core–shell dimension has a significant impact on the SPR band of Au@Ag core–shell NPs, which is also highly penetrating to cyanide etching (Figure 4). This nanosensor illustrates the linear cyanide concentration range of about 0.4–100 μM with an LOD of 0.4 μM. These Au@Ag core–shell NPs are self-possessed in agarose gels as transportable “test strips” (Figure 4c). The potential of these test strips is for cyanide quantitative detection as well as semi-colorimetric quantitative analysis. However, the author made it clear that this gel-based reaction happened considerably more slowly than it did in the solution medium. This nanosensor favors the specific detection of cyanide anions against other inorganic anions. This may be due to cyanide’s slower rate of dispersion in these test strips. Using Ag@Au core–shell NPs, the same research team [79] developed cyanide anion colorimetric detection. Ag@Au core–shell NPs demonstrate an excellent linear connection between the decreased SPR band at 520 nm and cyanide concentrations between 0.4 and 32 μM, with a detection limit of 0.16 μM. These core–shell NPs were functionalized with PS-40, and the PS 40-Ag@Au NPs had great salt stability, making them useful for the detection of cyanide in sewage water samples using complex media.

Successful Au@Au-Ag yolk–shell NP synthesis and colorimetry sensing for the cyanide anions in a complex matrix was accomplished by Zeng’s research group [84]. According to the authors, this technique had a quick reaction time and reached equilibrium in less than 3 min, which aided in proper functioning. Additionally, a computer program and smartphone app were also designed to provide accurate and timely results information for the determination of cyanide anions. To quickly monitor the cyanide in an aqueous medium, Zhang and co-workers [85] created a colorimetric nanoprobe based on Au@Ag core/shell NRs. When plasmonic metals were suspended in a colloidal solution containing cyanide, the aspect ratio of Au@Ag core/shell NRs was changed. This caused the SPR band to weaken and shift hypsochromically, and the color of the NRs also changed. In contrast to the spherical core–shell NPs-based technology, the current core/shell NRs-based nanosensor completes the reaction in less than a minute, demonstrating the efficiency of this approach. The ability to measure the amount of cyanide anion in ambient water samples was added to this optical nanosensor. The core–to–shell dimensional ratio of the plasmonic core–shell NPs was altered by successively etching the shells and cores with cyanide ions, which significantly changed the color along with the SPR band and made ultrasensitive visual sensors easier to use [86].

### 2.5. Anisotropic Plasmonic Nanomaterials-Based Colorimetric Sensors for Cyanide Ions

Anisotropic plasmonic nanomaterials (nanorods, nanoplates, nanotubes, nanocubes, nano boxes, and nanopyramids) are considered more effective for naked eye-based visual sensors than their isotropic plasmonic counterparts, due to advantages such as a broadly adjustable spectral peak of the SPR absorbance and controllability of color contrast [87]. These advantages allow for the development of naked-eye visual techniques that can provide a larger detection range of analyte concentration, superior sensitivity in analyte sensing, and visual determination based on higher color contrast. Gold nanorods (AuNRs) were developed by Lee’s research team [88] based on feasible colorimetric nano-assays for the detection of cyanide anions. This method is dependent on the SPR band analysis, which heavily depends on the aspect ratio of AuNRs. Additionally, the diagonal faces of AuNRs are selectively etched by the cyanide anion, which results in a reduction in their size–to–height ratio, a corresponding blue shift in an SPR band, and a color change from peacock blue to pink. This nanosensor demonstrates a cyanide detection range of 1.65 nM to 0.5 mM, with a detection limit of around 0.5 nM. Additionally, this approach was expanded for use with natural water samples.

The novel cyanide sensing optical technique described by Wang and co-workers [89] is based on unmodified Ag–Au alloy nano boxes that etch in the presence of cyanide anion, giving an SPR frequency shift associated with the analyte concentration. This technique demonstrated that local cyanide dosage alterations in HeLa cells and zebrafish embryos may be detected with nanomolar sensitivity. This sensor also demonstrates that use of Au-Ag nano-box enabled high sensitivity of a reliable sensing device for mapping biogenic and anthropogenic cyanide contaminations in vivo and in vitro.

Sasikumar and Ilanchelian [90] established a specific colorimetric methodology for chemosensory detection of cyanide anions using gold nanobipyramids (AuNBPs). With increasing cyanide anion addition, the optical density affecting the AuNBPs’ longitudinal LSPR peak drastically dropped with a hypsochromic shift. The low concentration of cyanide anion may also be detected with the naked eye using these absorbance-spectrum fluctuations along with a visible color change from wine red to pale red. This sensing strategy exhibits high specificity for detecting cyanide anions over other inorganic anions. Additionally, this approach was extended to include the use of cyanide anions in actual water samples. Anisotropic plasmonic materials-based colorimetric sensors for cyanide anions also follow the Elsner reaction-based sensing mechanism [87,88,89,90].

### 2.6. The Colorimetric Sensors for Cyanide Ions Using Plasmonic Nanoparticles with Peroxidase Activity

The peroxidase-like activity could combine with plasmonic nanoparticles to form reliable and effective chemosensors [91]. Based on analyte-induced shielding of the peroxidase-like activity of cysteamine-decorated AuNPs, Dan et al. [92] built a sensitive and targeted colorimetric platform for cyanide anions. The 3,3′,5,5′-tetramethylbenzidine (TMB) was oxidized by H_2_O_2_ with the cysteamine-decorated AuNPs in the presence of cyanide anions, providing blue-colored reactive oxidized TMB (oxTMB) with an absorbance maximum at the wavelength of 652 nm. The quantitative sensing of cyanide anions was enabled by detecting the changes in the absorption maxima at 652 nm. This nanosensor relied on an Elsner reaction-based sensing mechanism, with a limit of detection of 0.33 μM. This technique could apply to real water samples for quantitative sensing of cyanide ions.

**Table 1 nanomaterials-13-00290-t001:** Analytical merits of plasmonic metal nanoparticles based colorimetric sensors.

Materials	Advantages	Disadvantages	LOD	Real Samples	Ref.
SDS-AgNPs	Simple fabrication Highly selective	---	1.8 μM	Dam water	[50]
ESNPs	Merged with nanofiber	Less selective 2-mercaptobenzothiazole is interfering	0.46 μM	Pond, tap, and industrial water	[51]
Agrose-AgNPs	Highly selective	Short linear range	0.69 μM	Sea and river waters	[52]
Seaweed-AgNPs	Highly selective Merged with test strips	Highly selective merged with test strips	1 μM	River water	[53]
Rosmarinic Acid-AgNPs	Merged with agarose test strips	Iodine is interfering	0.01 μM	Tap and drinking water	[54]
Photochemical AgNPs	Highly sensitive Fluorescence-based detection.	Sulfide is interfering	2 μM	Pond and river water	[55]
TX-100-AuNPs	Highly selective Compared with traditional cyanide sensors	No portable usage	0.15 μM	Wastewater	[69]
ATP-AuNPs	Highly selective	The analytical application was not done. This method worked on specific pH only. 30 min incubation time	14 μM	---	[70]
PS-40 AuNPs	Highly selective	No portable usage	0.5 μM	Water samples, Cassava roots	[71]
Chitosan-AuNPs	Highly selective Portable usage	Less detection limit	2.3 μM	Water and blood samples	[72]
β-CD AuNPs	Highly selective Portable usage Merged with cotton swab Rapid	---	93 nM	Real water samples	[73]
PDA-AuNPs	Highly selective Wide linear range	pH selective	4.6 μM	Water samples	[74]
PVA-chitosan AuNPs	Highly selective Thin film Portable usage	Time-consuming for thin film making No analytical application	0.1 μM	---	[75]
Citrate-AuNPs	Highly selective Portable usage Merged with filter paper	---	7.68 μM	Tap and creek water samples	[76]
ECNPs-NC film-CuNPs	Highly selective Portable usage Cellulose fiber	Time-consuming for cellulose fiber making	0.58 μM	Water samples	[81]
Au@Ag core–shell NPs	Highly selective Portable usage Merged with test strips	---	0.4 μM	Tap, sea, lake, and industrial water samples	[83]
PS 40-Ag@Au core–shell NPs	Highly selective	No portable usage	0.16 μM	Drinking water	[79]
Au@Au–Ag yolk-shell NPs	Highly selective Rapid Merged with smartphone and computer programs.	Complicated synthetic procedure for NPs	---	Tap and bond water	[84]
Au@Ag core/shell NRs	Rapid Highly selective	No portable usage Complicated synthetic procedure for NPs	0.5 μM	---	[85]
AuNRs	Highly selective	No portable usage	0.5 nM	Tap, pond, and wastewater	[88]
Au-Ag nanoboxes	Highly selective Wide range of Applications	A skilled person was need for NPs synthesis	1 nM	Cell line detection	[89]
AuNBPs	Highly selective	No portable usage	1.58 nM	Tap, drinking, and seawater	[90]
Cysteamine- AuNPs	Highly selective	No portable usage External reagents need for this tactic	0.33 μM	Real water samples	[92]

### 2.7. Noble Metal Nanomaterials-Based Rayleigh Scattering Sensors for Cyanide Ions

Rayleigh scattering (RS) is an elastic scattering of light with scatterers (e.g., molecules) of a size much smaller than the wavelength of incident light. The RS results in scattered light with a wavelength the same as the incident wavelength [93,94]. Light-induced resonant oscillation of electrons on the metallic nanostructure surface also scatters light in the RS manner, leading to resonant absorbance of incident light energy at the scattering wavelengths, so-called resonant absorbance, such as that found in the SPR of noble metal nanoparticle surfaces [37,39,94,95,96]. 

T. Madrakian research group [97] reported extreme-trace cyanide detection with high specificity and sensitivity using SPR-based light scattering with AgNP-doped magnetic nanoparticles (Ag/Fe_3_O_4_). This composite nanoprobe reacted quickly with cyanide ions, leading to reduction in the RS intensity. This enabled cyanide concentration to be detected quantitively by measuring reduced RS intensity in a certain range and this technique could be applicable to spiked water. Moreover, Lionberger et al. [98] synthesized the three different sizes (14, 40, and 80 nm) of polysorbate 40 stabilized-AuNPs and applied the colorimetric and RS scattering-based sensing of cyanide anions in real water samples. The nanoprobe with 40 nm size turned out to be the most sensitive for cyanide sensing. This nanoprobe, integrated with a portable instrument, could be suitable as a field-deployable, fast diagnostic method.

## 3. Noble Metal Nanomaterial-Based Fluorescence Sensors

The second pattern of the optical approach is the spectrofluorometric technique, which has several advantages over other methods, including high selectivity and sensitivity, reduced sensing time, adaptability for online monitoring, and ease of sensing operation. This technique was reported to identify attention-required substances, such as drugs and chemicals [30,50]. It is necessary to select the specific substance utilized as a luminous material and to set the procedures for monitoring changes in the fluorescence spectrum. These two issues were the main challenges to fluorescence sensing technologies [99,100]. For the detection of dangerous substances and pharmaceuticals, several researchers have extensively employed organic compounds and dyes as fluorophores. Organic fluorophores dissolvable only by organic solvents are not suitable for practical sample analysis as they are harmful to the environment and humans. They also suffer from other drawbacks such as photobleaching, autofluorescence, spectrally broad emission with red tailoring, and narrow excitation spectra [101,102,103]. To solve these problems, size-modulated nanostructured materials such as metal nanoclusters (MNC) were used as fluorescence-based chemosensors. Advantages of the MNC, the fluorescent metal nanostructure, included superior photostability, low toxicity, and ultrafine size, making them alternatives to conventional fluorophores for detecting cyanide anions [50,104,105]. 

### 3.1. AgNC-Based Fluorescence Sensors for Cyanide Ions

Atomically specific MNCs are extremely small units with core sizes lower than 2 nm, and they are in between the plasmonic MNPs and the atomic regime [106]. Such MNCs exhibit incredibly unique optical and electrical properties, such as a highly emissive feature, molecule-like energy structure, and tremendous catalytic activity [107]. Due to their extraordinary physiochemical characteristics, such as their ultra-small size and excellent emissive qualities, AgNCs stand out among the noble MNCs. Such characteristics provide reliable optical methods for producing fluorophores for chemo-sensing and bioimaging applications [108]. Deoxyribonucleic acid (DNA)-stabilized AgNCs were described by Peng et al. [109] as a fluorescent probe for the precise and targeted detection of cyanide anions in natural water samples. The emissive character of DNA-AgNCs may be seriously affected by the presence of cyanide anion. The Elsner reaction with ground state complex formation serves as the sensing mechanism for this fluorescent module, which can detect the cyanide anion in a linear range of 0.10 to 0.35 μM with an LOD of 25.6 nM. 

### 3.2. AuNC-Based Fluorescence Sensors for Cyanide Ions

The specialized forms of gold nanomaterials known as photoluminescent AuNCs or gold nanodots (AuNDs), with diameters less than 3 nm, did not exhibit an SPR absorbance band in the visible region but exhibit photoluminescence in the viewable to near-infrared (NIR) area [110]. AuNCs have been used to develop intriguing optical chemo-sensing and imaging nanomaterials with a wide Stokes shift, prolonged luminescence lifetime, good photostability, biocompatibility, and acceptable stability [111]. Liu et al. [112] showed for the first time a red-emissive nanosensor based on BSA-stabilized AuNCs for exceptionally sensitive and selective cyanide anion quantification in a water-only medium. This technique focused on the emission quenching of the AuNCs caused by cyanide anion etching. This nanosensor’s remarkable specificity was due to the fact that other inorganic anions did not scratch the BSA-AuNC surface. Various naturally occurring water samples, such as tap water, groundwater, lake water, and pond water, that were spiked with cyanide anions were evaluated using this detection method.

Dong et al. [113] reported a straightforward fluorescence assay for the very precise cyanide sensing of lysozyme-templated AuNCs. The lysozyme-AuNCs developed using this approach have an average size of around 4 nm and exhibit a prominent red color emission peak at 650 nm when excited at 370 nm. The emissive properties of lysozyme-AuNCs were linearly diminished by cyanide ion concentrations in the range of 5 to 120 μM with an LOD of 0.19 μM. The same research team [114] published a study on the innovative and ecologically secure L-amino acid oxidase-capped AuNCs with red fluorescence for the detection of cyanide anions. The authors observed a notable emission quenching at 630 nm (excitation at 510 nm) in AuNCs in the presence of the cyanide anion. With an LOD of 0.18 μM, this fluorescent sensor provided two linear correlations between cyanide concentrations and AuNC emission intensities; for instance, in the ranges of 3.2 to 34 μM and 38.1 to 104 μM. 

The fabrication of innovative, highly photoluminescent trithiocyanuric acid AuNDs as the cyanide-detecting probe was reported by Vasimalai et al. [115]. The authors state that this method overcomes limitations of other preparative protocols in terms of cost, difficulty, timing, and environmental risk, and enables the growth of extremely potent luminescent AuNDs in less than 10 min at ambient conditions, with emission maxima at 623 nm and a considerable Stokes shift (213 nm). The emission nature of the synthesized AuNDs was suppressed and a significant hypsochromic shift was established by gradually increasing the cyanide concentration in a colloidal dispersion. This chemosensor showed a well-fitted linear connection for cyanide concentrations between 0.29 and 8.87 μM with an LOD of 150 nM. The ovalbumin-stabilized AuNCs were described as cyanide-selective colorimetric and fluorescence probes by the Ilanchelian research group [116]. In this method, the authors modify the surface functionalizing protein ovalbumin instead of lysozyme with the method described above [107]. This nanosensor exhibits excellent sensitivity (Figure 5a) with specificity for cyanide detection against other inorganic anions. (Figure 5b). Ovalbumin was also fairly economical compared with other proteins, such as lysozyme and bovine serum albumin. 

Based on Yang et al. [117], a nanosensor made of double-emissive AuNCs (DE-AuNCs) supported by hyperbranched polyethyleneimine was designed for ratiometric detection of cyanide ions via outward valency-state-driving etching. The red-fluorescence AuNCs with a high surface Au(I) content can be easily etched by cyanide anion, but the blue-fluorescence AuNCs with a nearly neutral charm can resist cyanide. As a consequence, this ratiometric sensor displays an emissive color change with an LOD of 10 nM. By measuring the amount of cyanide in urine and river water samples, this DE-AuNC-based fluorescent platform’s analytical potential was examined.

Hu et al. [118] described the ratiometric emission method for cyanide determination by blue-emissive carbon dots (CDs) and bright-red-emissive AuNCs. The two fluorescent nanoprobes were electrospun onto a nanofibrous membrane to provide the core–shell assembly of CDs/AuNCs-polyvinyl alcohol@cellulose for the on-site measurement of this method. The ratiometric fluctuation in the emissive nature of nanofiber resulted from the red emission of AuNCs being diminished, while the blue emissive nature of CDs remained unaffected upon chemical interaction with cyanide. This ratiometric nanosensor provided a cyanide LOD of 0.15 μM, which is significantly less than the WHO norms. The cost-effective, simple fabrication and emerging utility of CDs/AuNCs-polyvinylalcohol@cellulose nanofiber-based evaluation strips were emphasized. They are also useful in the early stages of cyanide anion sensing analysis in water samples. According to the previously stated idea [118], Wang et al. [119] developed the ratiometric fluorescence probe for the detection of cyanide anions. The Elsner reaction (Figure 6) is the primary mechanism used in the majority of AuNC-based fluorescence sensing of cyanide anion quantification [120,121,122]. In general, the results of HR-TEM were utilized to show that the surface of AuNCs had been etched by cyanide anion, and mass spectrometry was used to confirm the formation of Au(CN)_2_^−^. The cyanide sensors based on fluorescent metal nanoclusters and their detecting mechanisms are displayed in Table 2. 

### 3.3. Copper Nanocluster-Based Fluorescence Sensors for Cyanide Ions

Recent years have shown a lot of attention paid primarily to the distinct photoluminescent characteristics of CuNCs. Agreeable physiochemical characteristics of CuNCs with a core size of less than 2 nm connect CuNPs to atomic and molecular features [31,98,123]. CuNCs still seem to have significant problems, namely rapid oxidation, and a low quantum yield [31,104]. According to Safieh Momeni et al. [124], blue-colored emissive CuNPs can be used as a fluorophore for the very sensitive detection of the cyanide anion over other inorganic anions. This work emphasizes the high-yield production of blue emissive CuNPs with ascorbic acid as a protecting, reducing, and stabilizing agent without any extra reagents. Additionally, the presence of cyanide significantly reduced the luminous properties of CuNPs, which was a result of the strong chemical interaction between cyanide and CuNPs. This nanosensor showed a decent linear association of cyanide content in the range of 0.5–18 μM with an LOD of 0.37 μM, and this method was successfully used to quantify the cyanide anion in actual water samples.

By employing target-triggered emission quenching of thiosalicylic acid stabilized-CuNCs nanoassay, Jinshun Cang and colleagues [125] developed the double-sensing assay for the measurement of cyanide and nitrite anions. Due to the pH sensitivity of this optical chemosensor, nitrite detection was carried out in an acidic medium, whereas cyanide detection was done in an alkaline medium as the cyanide etching occurred more effectively in an alkaline medium than an acidic one. This technique was successfully applied to analyze samples of lake water. Fei Sun’s research group [126] reported on CuNCs with and without salicylaldehyde molecules for multi-cations and toxic cyanide anion sensing in real water samples and biological fluids. With rising quantities of cyanide anions, the fluorescence properties of CuNCs are routinely improved in this sensing technique. In this case, the aldehyde group was responsible for the nucleophilic attack of cyanide. As a result, CuNCs’ emissive properties were improved, and this approach showed an LOD of 0.51 M. This nanosensor was further developed for biological applications, such as for detecting cyanide ions in cancer cells. CuNC-based cyanide sensing techniques also follow the mechanism of metal-cyano complex production by cyanide etching of NC surfaces [124,125,126].

### 3.4. Bimetallic Nanocluster-Based Fluorescence Sensors for Cyanide Ions

Recently, bimetallic nanoclusters have gained more attention than single metallic nanoclusters due to their composition-dependent properties. Study on bi-metallic NCs has gained public awareness because of the improved optical characteristics, stability, biocompatibility, and photostability. Surprisingly, different bi-metallic NC descriptions continue to be more effective for chemosensors, bio-imaging, and drug delivery [127,128]. Lu Tian et al. [129] fabricated the bright-orange color emissive Au/Ag bimetallic NCs for cyanide anion sensing in real water samples. In this method, Au/Ag bimetallic NCs were produced in an egg-white albumin matrix with the use of a microwave. These bright Au/Ag bimetallic NC-based sensors produced an even more astonishingly robust reaction to the cyanide anion with quick, exact, and ultra-sensitive properties, and it adheres to the Elsner reaction-based sensing mechanism. The authors assert that their findings demonstrate the environmental friendliness and wide application potential of these Au/Ag bimetallic NCs for emerging applications, such as the measurement of bioimaging and environmental contamination.

**Table 2 nanomaterials-13-00290-t002:** Analytical parameters of metal nanoclusters-based fluorescence sensors.

Materials	Sensing Mechanism	Linear Range	LOD	Real Samples	Ref.
DNA-AgNCs	Fluorescence quenching of DNA-Ag NCs was a static fluorescence quenching caused by the interaction of cyanide	0.10–0.35 μM	25.6 nM	River water	[109]
BSA-AuNCs	Elsner reaction-based emission quenching	0.20–9.6 μM	200 nM	Ground, tap, pond, and lake water samples	[112]
Lysozyme-AuNCs	Elsner reaction-based emission quenching	5–120 μM	190 nM	---	[113]
L-Aminoacid-AuNCs	Elsner reaction-based emission quenching	2.3–34 μM	180 nM	River and tap water samples	[114]
AuNDs	Elsner reaction-based emission quenching	0.29–8.87 μM	150 nM	Natural water samples	[115]
Ovalbumin-AuNCs	Elsner reaction-based emission quenching	0.5–7.5 μM	68 nM	Tap, drinking, and dam water samples	[116]
DE-Au NCs	Cyanide etching of AuNCs surface	0.02–1 μM	10 nM	Water and urine samples	[117]
CDs/AuNCs-polyvinylalcohol@ cellulose	Elsner reaction-based ratiometric emission quenching	0.2–20 μM	0.15 μM	Tap water	[118]
CDs-AuNCs	Elsner reaction-based ratiometric emission quenching	12.5–75 μM	---	Food and drink samples	[119]
Lysozyme-NP-AuNCs	Elsner reaction-based ratiometric emission quenching	3–100 μM	1 μM	Tap water and soil	[120]
BSA-Ce^3+^-AuNCs	Elsner reaction-based ratiometric emission quenching	0.1–15 μM	50 nM	Drinking and pond water samples	[121]
CuNPs	Metal-cyano complex formation and strong interaction between nanoprobe and analyte	0.5–18 μM	0.37 μM	River water	[124]
Thiosalicylic acid -CuNCs	Metal-cyano complex formation and strong interaction between nanoprobe and analyte	0.01–1 μM	5 nM	Lake water	[125]
Salicylaldehyde-CuNCs	Nucleophilic addition of salicylaldehyde groups in CuNCs by cyanide	---	0.51 μM	Bio-imaging	[126]
Au/Ag bimettalic NCs	Elsner reaction-based emission quenching	0.5–50 μM	138 nM	Real water samples and live cell imaging	[129]

### 3.5. Fluorescence Using Fluorophores Coupled with Plasmonic Nanoparticles for Cyanide Ions

The combination of fluorophores (organic/inorganic molecules, carbon dots, semiconductor quantum dots (QDs) with graphene quantum dots and plasmonic NPs has emerged as one of the most exciting platforms for fluorometry-based sensors. This combination can produce a new type of composite fluorescent material with a distinct structure and excellent photochemical and photophysical properties [130,131]. Table 3 provides a summary of these remarkable optical responses that enable the development of a novel class of optical chemosensors for cyanide anion. Li Shang et al. [132] reported a highly specific and sensitive luminescent tactic for sensing cyanide with the support of luminophore-coupled NPs. Rhodamine B is the prototype luminophore used in this approach because it is highly photostable, water-soluble, fluorescent, and positively charged. With a high FRET process, rhodamine B may have electrostatically connected with the negatively charged citrate-capped AuNPs. The efficient FRET and the IFE mechanism between rhodamine B-functionalized AuNPs result in their weak fluorescence. AuNPs were progressively dissolved in the presence of cyanide anions, and as a result, rhodamine B’s emission nature was restored and the damage caused by the AuNPs was identified. Cyanide anions were then quantified from this emission recovery. This chemosensor has an 80 nM LOD and a linear range of cyanide concentration from 0.15 to 45 μM. Several research groups have developed fluorescence detection of cyanide by various luminophore (organic/inorganic molecules) functionalized plasmonic NPs [133,134,135,136,137] in accordance with the principle outlined above [132].

Rezaei research group [138] fabricated the fluorescence turn-on cyanide detection based on semiconductor cadmium telluride (CdTe) QD-coupled AgNPs as a fluorophore. This nanosensor was also found to operate in a ‘turn-on’ manner, which is typically more sensitive than a ‘turn-off’ pathway. Cyanide concentrations may be found linearly ranging from 0.01 to 2.5 μg/mL in CdTe QD-functionalized AgNP-based experiments. Additionally, this technique was used to analyze samples of human serum and wastewater. For the purpose of detecting and visualizing endogenic cyanide anions, Lili Wang et al. [139] demonstrated a simple nanosensor employing graphene quantum dots conjugated with AuNPs. This cyanide identification technique may attain an LOD of 0.52 μM without any interference from associated indications from a biological matrix, thanks to the fluorescence quenching efficacy of two-photon GQDs with AuNPs. This nanoprobe also found cyanide in plant tissues, which can be investigated using the bio-imaging method. The highly emissive nitrogen-doped carbon dots with plasmonic MNPs (Ag and AuNPs)-involved detector for cyanide anions based on the IFE technique were described by Zhang and coworkers [140]. Surface of the MNPs can be etched by the addition of cyanide anions to fluorophore-conjugated MNPs, resulting in a decrease in absorbance and the restoration of the IFE-reduced fluorescence. This method produced an LOD of 2 μM of cyanide with the measurement time of 20 min in actual water samples. Majority of the strategies for fluorescence detection of cyanide anions using fluorophore-functionalized plasmonic MNP used a similar IFE/FRET quenching process, from which metal-cyano formation produces fluorescence turn-on by an emission recovery mechanism [132,133,134,135,136,137,138,139,140,141,142] (Figure 7). 

**Table 3 nanomaterials-13-00290-t003:** Analytical factors of fluorophore-functionalized plasmonic metal nanoparticles.

Materials	Sensing Mechanism	Linear Range	LOD	Real Samples	Ref.
Rhodamine B-AuNPs	AuNPs made IFE process-based emission quenching-cyanide etching AuNPs surface followed by fluorescence recovery	0.15–45 μM	80 nM	---	[132]
PF-AgNPs	AgNPs made IFE process-based emission quenching-cyanide etching AgNPs surface followed by fluorescence recovery	0.5–600 μM	0.25 μM	Tap water	[133]
Polyfluorene with AuNPs	Fluorescence of polymer quenched by AuNPs turned on, then the more stable Au(CN)_2_^−^ were formed	0.05–130 μM	---	Groundwater, tap water, boiled water, and lake water samples	[134]
Polyacetylene-AuNPs	Fluorescence of polymer quenched by Au NPs turned on, then the more stable Au(CN)_2_^−^ were formed	---	---	Groundwater, tap water, boiled water, and lake water	[135]
BSA-FITC-Au NPs	BSA-AuNPs made IFE process-based emission quenching-cyanide etching AuNPs surface followed by fluorescence recovery	0–10 μM	1 μM	Pond, sea, and tap water samples	[136]
FITC-PS-40-Au NPs	PS-AuNPs made IFE process-based emission quenching-cyanide etching AuNPs surface followed by fluorescence recovery	0–50 μM	0.1 μM	Tap, river, drinking, and seawater samples	[137]
CdTe QDs-AgNPs	Fluorescence of QDs quenched by AgNPs turned on, then the more stable Ag(CN)_2_^−^ were formed	0.38–96 μM	0.15 nM	Serum and wastewater samples	[138]
GQDs-AuNPs	AuNPs made FRET process-based emission quenching-cyanide etching AuNPs surface followed by fluorescence recovery	1–200 μM	0.52 μM	Plant tissues	[139]
CDs-Au and AgNPs	MNPs surface can be etched by cyanide, bringing on absorbance decrease and regenerating the IFE-reduced fluorescence	1–100 μM	2 μM	Serum and water samples	[140]
N,S, GQDs-AgNPs	AgNPs made IFE process-based emission quenching-cyanide etching AgNPs surface followed by fluorescence recovery	10–500 μM	0.52 μM	Tap water samples	[141]
Fluorophore- DNA CuNPs	Nano lamp was constructed on the basis of the optical interaction between CuNPs and the fluorophore and the highly effective etching effect of cyanide on CuNPs	2.5–20 μM	1.96 μM	Live cell imaging	[142]

## 4. Noble Metal Nanomaterials-Based SERS Sensors for Cyanide Ions

The SERS effect results from the use of the appropriate MNPs placed in proximity to Raman active molecules (ions) under an excitation light source [143]. Typical SERS substrates have the roughened surface made of metals such as Ag/Cu/Au. The SERS technique requires the adsorption of the analyte molecules onto the SERS substrate. Upon adsorption onto the SERS surface, the Raman signal of the analyte is enhanced, and the resultant signal intensity is comparable to that obtained by fluorescence. To detect the cyanide anion, many researchers have coated the surface of the SERS substrate with the plasmonic MNPs and applied them as a SERS probe [143,144,145,146]. D. Senapati et al. [147] reported an AuNP-based SERS nanosensor for cyanide ion detection at the level of parts per trillion and for studying cyanide anion–AuNPs interaction. After the mixing of cyanide anion with AuNPs in 2 min, the authors clearly observed three bands in the SERS data: an Au-C stretching frequency around -370 cm^−1^ owing to the construction of Au-C≡N bond; an Au-C≡N bending frequency around ~300 cm^−1^; and a solid Raman peak at 2154 cm^−1^, attributed to the pure C≡N stretching frequency [148]. The SERS peak around 2154 cm^−1^ was enhanced after the addition of cyanide anions, which is due to the well-known Elsner reaction. Furthermore, this plasmonic nanomaterials-based SERS nanoprobe was utilized to quantify cyanide concentration in real water samples (Figure 8). Based on this principle, many research groups developed the AuNP-based SERS probe and applied the cyanide determination in water [149,150]. 

Z. Cheng’s research group [151] fabricated the Ag nanoplate as a SERS probe for detecting cyanide. This nanoarray was constructed by employing an electro-deposition technique with a reduced current density. Such a nano-array was an excellent SERS substratum, with structural stability, repeatability, and good activity because of the sphere-shaped structure. After adding cyanide anions, the signal of the SERS peak was enhanced at around ~2140 cm^−1^ corresponding to C≡N. It was possible to calculate the quantity of cyanide anions using these increased parameters. The LOD detected by this nanosensor was 0.1 parts per billion. Several research teams have reported using different molecular-functionalized AgNPs as SERS substrates for cyanide detection at trace levels [152,153]. 

## 5. Conclusions and Outlooks

In this review, we concentrated on the previously reported optical sensing techniques based on nanostructured noble metal materials for detecting cyanide ions. The four different types of nanosensors, SPR-based colorimetric probes, spectrofluorimetric methods, SERS, and RS nanosensors, have been assembled. The real-time application capacity of the noble metal nanomaterials-based sensing techniques in environmental water samples has been identified. These methods enabled the cyanide ion detection with super-specificity, ultra-sensitivity in a simple format.

Noble metal nanomaterials-based cyanide sensors are especially useful, since it might be difficult to access modern facilities with cutting-edge technology and qualified workers. Until now, optical sensors used in this field have mostly relied on morphological alterations and the subsequently induced color changes. Although metal nanostructured materials-based sensors provide a variety of advantages for quantifying cyanide ions, they still need to be further developed and overcome the challenges, namely, as in the following: ❖The very active characteristics and poor self-stability of metal nanostructured materials frequently limit the functionality of complex actual systems, making the nanosensors more proof-of-concept devices, especially for the aggregation-based colorimetric sensing technique. ❖Scientists should give special consideration to synthesizing novel, highly stable noble metal nanostructured materials and to developing new modification techniques to expand the functionality and analytical usage for real samples to satisfy the determination requirements for cyanide ions in problematic environments, such as wastewater, seawater, biological samples, and food additives. ❖Another key aspect of nanosensors is specificity. To effectively boost sensing specificity, development strategies should use the proper surface-functionalizing ligands, as well as designing new, rapid ligands with high selectivity for cyanide ions. ❖The next challenge is always the rapid on-site detection of cyanide ions. For quicker and more effective devices, with profitable industrial applications for cyanide anion monitoring, useful nanosensors should be integrated and combined with test strips, cotton swaps, gels, microfluidic/paper chips, membranes, smartphones, image processing techniques, and other technologies and strategies. 

## Data Availability

The data presented in this study are available on request from the corresponding author.

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
