# Peer review of "Optical Sensing of Toxic Cyanide Anions Using Noble Metal Nanomaterials"

_nanomaterials, 2023, doi:10.3390/nano13020290_

Round 1

Reviewer 1 Report

Referee Report on the paper «Optical sensing of toxic cyanide anions using noble metal nanomaterials” by R. Rajamanikandan, K. Sasikumar and H. Ju, submitted to Nanomaterials.

In this review paper, the authors analyze analytical performance and merits of different noble metal nanomaterials (nanoparticles, core-shell nanoparticles, nanorods, bimetallic nanoclusters, etc.), and different implying them methods (colorimetric, fluorescent), to detect toxic cyanide metal ions in water. The paper contains a lot of useful and timely information, extensive and reasonably full bibliography, not badly written and will be useful for the journal readers. I recommend acceptance. My comments are minor.

1.     The main shortcoming is the use of two different units for concentrations of ions when discussing the limits of detection: microgram/L (or microgram/mL) and microM; these different units are used even in the neighboring cells of the Tables! The relation between these units should be very clearly discussed in the beginning of the review.

2.     The text contains many abbreviations (probably unavoidable in such type of paper), but they are not always explained immediately after the first appearance. What are ROS (line 151), PVA-chitosan (line 222), PS-40 (line 265), BSA (line 397), IFE-FRET (lines 551 and 557)? Even LOD, evidently “limit of detection”, apparently is not explicitly explained.

Origin of some abbreviations are not clear, e.g. ESNPs - "embedded silver nanoparticles in transparent nanopaper" (ESNPs) according to ref. 61, or PDA- line 208: gold nanoparticles stabilized by poly (N,N-dimethylaminoethyl methacrylate) (PDMAEMA) (Au-PDA) according to Ref. 74.

3.     It would be useful to discuss the question of the stability of silver – based NP. How stable they are in comparison with gold-based and copper-based NP?

4.     In the conclusion, the authors state that the key aspect of the nanosensors is specificity, which is desirable to boost (line 616). This question almost was not touched in the main text. Is it possible to discuss the specificity at least for one-two nanosensors discussed?

Finally: the English is overall quite good. Still some editing seems would be useful. For example, what the following means “relatively could exhibit photoluminescence” (393), “Agreeable physiochemical characteristics of CuNCs with a core size of less than 2 nm connect CuNPs to atomic and molecular structures” (471), “The scientific community is not limited to single metallic NCs structures” (499)?  Or “is typically most

sensitive than a “turn-off” probe” (540)?

Author Response

Response Letter

Manuscript Id: nanomaterials-2146314

Manuscript title: Optical sensing of toxic cyanide anions using noble metal nanomaterials

Dear Editor,

The authors are grateful to the referee for the useful comments. We have thoughtfully taken into account these comments and modified the manuscript accordingly. The detailed corrections are listed below point by point:

Reviewers' comments:

Reviewer-1

In this review paper, the authors analyze analytical performance and merits of different noble metal nanomaterials (nanoparticles, core-shell nanoparticles, nanorods, bimetallic nanoclusters, etc.), and different implying them methods (colorimetric, fluorescent), to detect toxic cyanide metal ions in water. The paper contains a lot of useful and timely information, extensive and reasonably full bibliography, not badly written and will be useful for the journal readers. I recommend acceptance. My comments are minor.

We are thankful for your thorough evaluation of our work and you are recommending it for publication in this journal after a few corrections.

Comment 1.

The main shortcoming is the use of two different units for concentrations of ions when discussing the limits of detection: microgram/L (or microgram/mL) and microM; these different units are used even in the neighboring cells of the Tables! The relation between these units should be very clearly discussed in the beginning of the review.

Answer

As per the reviewer’s advice, we have made the necessary corrections in the revised manuscript.

Comment 2.

The text contains many abbreviations (probably unavoidable in such type of paper), but they are not always explained immediately after the first appearance. What are ROS (line 151), PVA-chitosan (line 222), PS-40 (line 265), BSA (line 397), IFE-FRET (lines 551 and 557)? Even LOD, evidently “limit of detection”, apparently is not explicitly explained. Origin of some abbreviations are not clear, e.g. ESNPs - "embedded silver nanoparticles in transparent nanopaper" (ESNPs) according to ref. 61, or PDA- line 208: gold nanoparticles stabilized by poly (N,N-dimethylaminoethyl methacrylate) (PDMAEMA) (Au-PDA) according to Ref. 74.

Answer

Thank you for this reviewer’s comment. We have made an abbreviation section and included it in the revised manuscript as per the reviewer’s feedback.

Comment 3.

It would be useful to discuss the question of the stability of silver – based NP. How stable they are in comparison with gold-based and copper-based NP?

Answer

As per the reviewer’s suggestion, we have made appropriate changes in the revised version of the manuscript.

Comment 4

In the conclusion, the authors state that the key aspect of the nanosensors is specificity, which is desirable to boost (line 616). This question almost was not touched in the main text. Is it possible to discuss the specificity at least for one-two nanosensors discussed?

Answer

We have made appropriate modifications to the revised manuscript in accordance with the reviewer's advice. (Figure 5b)

Comment 5

Finally: the English is overall quite good. Still some editing seems would be useful. For example, what the following means “relatively could exhibit photoluminescence” (393), “Agreeable physiochemical characteristics of CuNCs with a core size of less than 2 nm connect CuNPs to atomic and molecular structures” (471), “The scientific community is not limited to single metallic NCs structures” (499)?  Or “is typically most sensitive than a “turn-off” probe” (540)?

Answer

We have corrected the typographical and grammatical mistakes in the revised text in accordance with the reviewer's suggestions.

Reviewer-2

The review covers the noble metal nanomaterials-based sensors for cyanide ions detection. Several approaches considered, such as those based on colorimetry, fluorescence, Rayleigh scattering (RS), and surface-enhanced Raman scattering (SERS). Schematic representation for the optical chemosensors greatly facilitates the perception of the material. The authors give examples of recent works on each of the indicated types of measuring systems. Additionally, major challenges associated with these platforms are also addressed while future perspectives are given with the directions towards resolving these issues.

We respect and are grateful for this reviewer's appreciation of the work.

In general, the material is presented in an accessible language with a large amount of data obtained by various methods. The work can be recommended for publication after correcting minor flaws that can be eliminated during the preparation of the proofs.

Answer

In accordance with the reviewer’s comment, we have carried out the necessary changes in the revised version of the manuscript.

Reviewer 2 Report

The review covers the noble metal nanomaterials-based sensors for cyanide ions detection. Several approaches considered, such as those based on colorimetry, fluorescence, Rayleigh scattering (RS), and surface-enhanced Raman scattering (SERS). Schematic representation for the optical chemosensors greatly facilitates the perception of the material. The authors give examples of recent works on each of the indicated types of measuring systems. Additionally, major challenges associated with these platforms are also addressed while future perspectives are given with the directions towards resolving these issues.

In general, the material is presented in an accessible language with a large amount of data obtained by various methods. The work can be recommended for publication after correcting minor flaws that can be eliminated during the preparation of the proofs.

Author Response

(The authors gave the same response as above.)
